# Pushing LLMs to Their Logical Reasoning Bound: The Role of Data Reasoning Intensity

## Abstract

Recent advances in large language models (LLMs) highlight the importance of training data structure and quality in shaping reasoning behavior. However, most existing approaches focus on transforming data formats while neglecting the internal reasoning complexity of training samples, leaving the reasoning potential of data underexplored and underutilized. In this work, we posit that LLM logical reasoning performance is jointly constrained by the potential of the training data and the cognitive capacity of the model. To make this relationship measurable, we introduce *Data Reasoning Intensity* (DRI), a novel metric that quantifies the latent logical reasoning complexity of samples by decomposing and aggregating their logical structures. This allows us to analyze how well current LLMs utilize logical reasoning signals and identify performance gaps relative to data potential. Based on this insight, we introduce a *re-cognizing optimization strategy* that systematically enhances the logical reasoning intensity of training data. Rather than increasing data volume, our method re-optimizes existing samples to better align with the LLM's logical reasoning bounder. Extensive experiments show that our approach significantly improves performance and generalization over data-centric strategies. We further validate our method under a reinforcement learning framework. Our results indicate that prioritizing reasoning complexity in data rather than sheer scale or superficial form is essential to realizing LLMs' full cognitive potential. Our code is available in the supplementary file.

## 1 Introduction

The reasoning ability of LLMs (OpenAI, 2023; DeepSeek-AI, 2024; Cheng et al., 2025; Chen et al., 2025d; Gao et al., 2025b; Ke et al., 2025; Zhou et al., 2024; Qiao et al., 2023) has emerged as a core metric for evaluating their cognitive alignment with human-like problem-solving. With the breakthrough of LLMs in logical reasoning tasks (Chen et al., 2025c; Li et al., 2025b; Feng et al., 2025; Wang et al., 2025a), optimizing the cognitive abilities of models from the perspective of training data has become the mainstream paradigm (Wu et al., 2025; Kandpal & Raffel, 2025; Peng et al., 2025; Prystawski et al., 2023; Chen et al., 2024; Kim et al., 2025). By reconstructing the cognitive expression form of the training data rather than simply expanding the data scale, it has demonstrated the crucial influence of data quality on the boundary of LLM's logical capabilities.

In this paper, we aim to explore LLM's logical reasoning ability from the data-centric perspective. Concretely, advanced methods such as DeepSeek-AI et al. (2025), indicate that it is not more data, but more complex and logically structured data that can better stimulate the reasoning potential of LLMs. Zhou et al. (2023); Ye et al. (2025) demonstrate that the structural guidance of training data is more effective than the volume of data in shaping the capabilities and behavioral patterns of LLMs. Indeed, the complexity of data can effectively stimulate and enhance the reasoning ability of LLMs only when the data has a reasonable structure, is close to the model's experience, and is near the boundary of its reasoning (Chen et al., 2024; Bai et al., 2025; Bi et al., 2024; Prystawski et al., 2023; Liu et al., 2025). Similarly, Peng et al. (2025) shows that the quality structure of training data plays a decisive role in reasoning ability, and precise data selection is better than simple data increment.

However, although existing research has made some progress, the current paradigm has fundamental bottlenecks: its optimization only stays at the superficial level of transforming the form of data

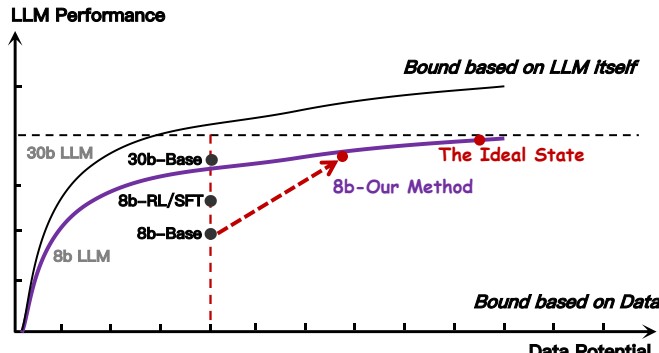

Figure 1: The illustrative visualization of data potential and LLM logical reasoning performance. The horizontal axis denotes data potential and the vertical axis denotes an LLM's logical-reasoning performance. Each curve corresponds to a specific LLM (e.g., different sizes) performance bound under a fixed architecture. As the potential of data increases, the performance of LLMs usually improves, but eventually it will reach an upper limit determined jointly by the model's capacity and the limitations of the data.

expression, while neglecting the in-depth exploration of the data's internal reasoning complexity. Specifically, existing methods face two major challenges:

- **Lack of efficient quantification:** Existing methods lack accurate measures of the logical reasoning complexity contained in data. For LLM's logical reasoning, logical nodes are crucial components, as they are closely related to the depth, breadth, and accuracy of the reasoning process. As a result, estimating the inherent reasoning complexity of the data itself becomes extremely challenging.

- **Blind analysis and optimization:** Although some complexity metrics do exist, few studies have analyzed the impact of training data from the perspective of the logical reasoning bounds of LLM. At the same time, how to leverage these findings in a reverse manner to guide training optimization and proactively enhance LLM's reasoning abilities remains a key challenge.

To make this gap measurable, we introduce **Data Reasoning Intensity (DRI)**, a practical metric that disassembles the reasoning pipeline into logically coherent steps and re-aggregates them into a single scalar that captures the complexity of the entire logical reasoning chain. Additionally, we suppose that the logical reasoning performance of LLMs is constrained by two bounds: the potential of the training data and the cognitive potential of the models themselves. As illustrated in Figure 1, current LLMs remain far from an ideal state in utilizing available reasoning data—indicating that they have not yet reached the ceiling of either data potential or cognitive capacity. By applying our proposed metric to the training corpora of logical-reasoning benchmarks, we observe measurable accuracy gains within the same dataset, confirming that the attainable performance ceiling has not been reached. Guided by these observations, we introduce a *re-cognizing optimization* strategy that systematically enhances the reasoning signals in training data, thereby pushing LLMs closer to their potential logical reasoning performance.

Our main contributions are as follows:

- We propose a novel metric for evaluating data reasoning potential and analyze the logical reasoning capabilities of LLMs from a data-centric perspective. Our analysis reveals that the training data still holds untapped potential and that current LLMs are far from reaching their performance ceiling.

- We propose a novel optimization approach, the *re-cognizing optimization* strategy, which reshapes and enhances the logical reasoning abilities of LLMs across diverse training samples, thereby pushing them closer to their reasoning bound.

- We validate the existence and rationality of our quantitative metric on logical reasoning tasks. Furthermore, we explore the effectiveness of our proposed optimization strategy, demonstrating its superiority over existing data-centric training paradigms. We also validate its efficacy under a reinforcement learning framework.

## 2 RELATED WORK

### 2.1 DATA-CENTRIC REASONING FOR LLMS

Recent work increasingly emphasizes that, beyond model architecture and algorithm design, the structure, provenance, and difficulty of training data play a critical role in shaping LLM reasoning performance (Chen et al., 2025c; Qu et al., 2025; Sui et al., 2025; Ruis et al., 2025). This has led to a data-centric shift in LLM development, where efforts focus on improving data composition and quality to enhance reasoning ability (Wu et al., 2025; Jin et al., 2024; Wang et al., 2025b). Many studies highlight that higher-quality, logically structured examples—rather than simply more data—yield better generalization and reasoning performance (Peng et al., 2025; Yu et al., 2025; Li et al., 2025a; Wettig et al., 2024; Zhao et al., 2024; Yu et al., 2024). In particular, aligning sample difficulty with model capability is identified as key to effective training (Gao et al., 2025a). Instruction tuning research supports this view, showing that both prompt quality and exposure timing influence reasoning emergence (Qingsong et al., 2025; Kim & Lee, 2024). Meanwhile, Kandpal & Raffel (2025) highlight the often-overlooked human labor cost in curating such training data.

### 2.2 REASONING EMERGENCE FOR LLMS

Complementary to these data-centric strategies, other studies investigate how reasoning capabilities emerge from the interaction between model cognition and structured input. Stepwise reasoning, for example, has been proposed as an emergent property of sequential data exposure rather than a fixed architectural feature (Prystawski et al., 2023). Furthermore, only data within a suitable complexity range appears to effectively stimulate reasoning behavior (Bi et al., 2024), suggesting that model performance is bounded by cognitive processing capacity. A number of empirical studies further show that small, carefully curated datasets often outperform larger but noisier corpora in supporting reasoning skills (Ma et al., 2025; Ye et al., 2025; Morishita et al., 2024; Wang et al., 2025c; Yang et al., 2025; Hua et al., 2025), reinforcing the value of reasoning supervision that is both selective and structurally rich. Prior work underscores data's role in LLM reasoning but often lacks precise difficulty metrics and clear goals. We introduce DRI, a unified score for reasoning potential, and *re-cognizing optimization*, which emphasizes high DRI examples while preserving diversity to enhance LLM's logical reasoning performance.

## 3 METHODOLOGY

Inspired by the Roofline Model (Williams et al., 2009; Cao et al., 2025), we frame LLM logical reasoning performance as an efficiency ratio between data-driven reasoning potential and model-intrinsic cognitive cost. For a model $\mathcal{M}$ evaluated on a dataset $\mathcal{D}$, the effective reasoning capability $\eta$ is defined as

$$\eta(\mathcal{M}, \mathcal{D}) = \frac{E(\mathcal{D})}{C(\mathcal{M})} \tag{1}$$

where $E(\mathcal{D})$ quantifies the latent logical reasoning demand encoded in the dataset—such as compositional complexity, multi-step inference depth, or symbolic abstraction—while $C(\mathcal{M})$ represents the model's intrinsic cognitive cost, encompassing architectural capacity, parameter scale, and reasoning FLOPs. Because $C(\mathcal{M})$ reflects hardware-bound and architecture-bound properties that are often costly or slow to modify, the most practical path to improving $\eta$ lies in enriching $E(\mathcal{D})$ through carefully designed, reasoning-intensive data.

Accordingly, we introduce the DRI score to quantify and maximize $E(\mathcal{D})$. It operates in two stages: first, we decompose each sample's structured reasoning trace into its logical components; second, we compute a DRI score that quantifies the reasoning potential embedded in that structured chain. By focusing on quantifying and raising $E(\mathcal{D})$, our approach unlocks latent training-data potential and drives enhanced LLM reasoning performance. Detailed definitions, explanations and proofs are provided in the Appendix E.

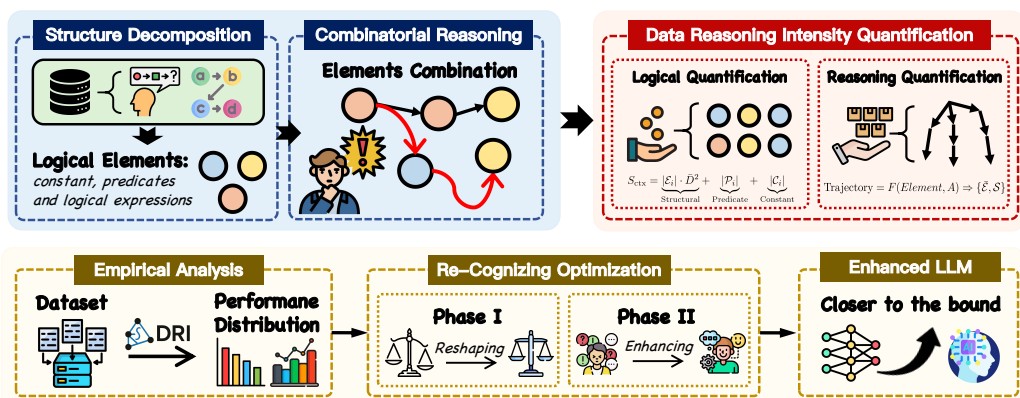

Figure 2: The overall framework. **Top:** We first extract logical elements from each example and perform combinatorial reasoning to derive a *Data Reasoning Intensity* (DRI) score. **Bottom:** We then analyze the performance distribution across DRI levels on multiple datasets. Based on this, we propose the *re-cognizing optimization* strategy: the first stage reshapes the model's recognition of reasoning patterns, and the second enhances its logical reasoning capability, thereby improving overall performance.

## 3.1 Decomposition and Combinatorial Reasoning

LLM logical reasoning begins as a collection of logical elements (Chen et al., 2025c), so we hypothesize that the reasoning process of LLMs involves structured reasoning. We first deconstruct the logical structure of the data and then conduct structural reasoning based on the deconstructed logical elements. The process begins by decomposing each question $Q$ into fundamental logical components via a distillation function $f$, implemented using LLMs such as DeepSeek-V3 or GPT-4o. We then extract the three core logical elements:

$$f(Q) \Rightarrow \{\mathcal{P}, \mathcal{C}, \mathcal{E}\} \tag{2}$$

where $\mathcal{C}$ is the constant set (e.g., "YoungBoy") and $\mathcal{P}$ is the predicates set capturing relationships (e.g., *IsYoung*(x)). $\mathcal{E}$ is the logic structure sets combining predicates and constants (such as first-order logic expressions).

Based on logical deconstruction, we need to perform combinatorial reasoning on the obtained logical elements. we define a combinatorial reasoning function $F$, implemented with an LLM, that leverages the distilled logical elements $\{\mathcal{P}, \mathcal{C}, \mathcal{E}\}$ and the candidate answer set $A$ to produce the precondition structure $\bar{\mathcal{E}}$ and the reasoning step sequence $\mathcal{S}$:

$$\text{Trajectory} = F(Element, A) \Rightarrow \{\bar{\mathcal{E}}, \mathcal{S}\} \tag{3}$$

where $\bar{\mathcal{E}}$ is the precondition logic structure and $\mathcal{S}$ is the single reasoning chain required for the reasoning trajectory. Detailedly, each reasoning node $s_k \in S$ contains:

$$s_k = (\#\text{Operations}_k, D_{\text{nest}}^k, \text{Expression}_k) \tag{4}$$

where $\#\text{Operations}_k$ captures the logical operators (AND/OR/NOT), $D_{\text{nest}}^k$ is the expression's nesting depth, and $\text{Expression}_k$ is its formal representation.

## 3.2 Data Reasoning Intensity

**Logical Intensity Quantification** Following logical decomposition, we compute a context score $S_{\text{ctx}}$ from three dimensions extracted from the question's logical elements:

$$S_{\text{ctx}} = \underbrace{|\mathcal{E}_i| \cdot \bar{D}^2}_{\text{Structural}} + \underbrace{|\mathcal{P}_i|}_{\text{Predicate}} + \underbrace{|\mathcal{C}_i|}_{\text{Constant}} \tag{5}$$

where $|\mathcal{E}_i|$ counts logical expressions, $\bar{D}$ is their average nesting depth (calculated via parse tree analysis), $|\mathcal{P}_i|$ and $|\mathcal{C}_i|$ tally unique predicates and constants respectively.

**Reasoning Intensity Quantification**  Building on combinatorial reasoning, each answer option's score $S_{\text{opt}}^{(l)}$ combines its precondition intensity with step-by-step deduction intensity:

$$S_{\text{opt}}^{(l)} = \underbrace{|\mathcal{R}_l| \cdot \bar{D}_l^2}_{\text{Preconditions}} + \sum_{k=1}^{T_l} \underbrace{\left(1 + \#\text{Operations}_{l,k}\right) D_{l,k}^2}_{\text{Step } k} \tag{6}$$

where $l \in \{1, \dots, L\}$ indexes the $L$ answer options, $\mathcal{R}_l$ is the set of precondition expressions for option $l$ with average nesting depth $\bar{D}_l$, and $T_l$ is the number of reasoning steps for option $l$. Each step $k$ contributes according to its operator count $\#\text{Operations}_{l,k}$ and nesting depth $D_{l,k}$.

Since decomposition and combinatorial reasoning metrics capture different facets of DRI and live on disparate scales, we first merge them into a raw intensity measure $S_{\text{raw}}$:

$$S_{\text{raw}} = S_{\text{ctx}} + \sum_{l=1}^{L} S_{\text{opt}}^{(l)} \tag{7}$$

To bound the resulting range within [0,1] and **obtain the final DRI score** $S$, we apply logarithmic compression followed by sigmoid normalization:

$$S = \sigma \left( \gamma \cdot \frac{\log(S_{\text{raw}} + 1) - \mu}{\sqrt{\sigma^2 + \epsilon}} + \beta \right) \tag{8}$$

where $\sigma$ denotes the sigmoid function, $\mu/\sigma^2$ are dataset statistics, and parameters ($\gamma = 1, \beta = 0, \epsilon = 10^{-5}$) ensure stable [0,1] normalization.

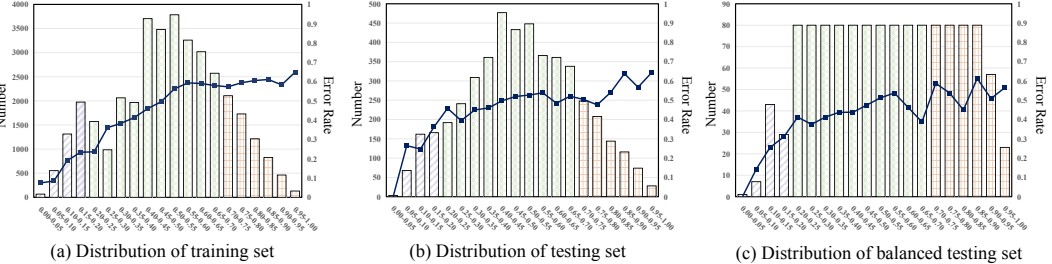

(a) Distribution of training set    (b) Distribution of testing set    (c) Distribution of balanced testing set

Figure 3: Effectiveness verification for DRI. Sample counts (bars, left axis) and model error rates (lines, right axis) are shown across DRI score bins. All three panels share the same layout: the x-axis divides the score range into uniform intervals, the bar height indicates the number of examples per interval, and the overlaid line traces the error rate. (a) Training-set distribution and error. (b) Original test-set distribution. (c) Balanced test-set distribution.

## 4 EXPERIMENTAL SETTINGS

In this section, we conduct experiments to address the following research questions:

- **RQ1**: Is the DRI score effective in enhancing the logical reasoning performance of LLMs?
- **RQ2**: How can DRI be effectively leveraged to improve LLMs' performance during training?
- **RQ3**: Is the *re-cognizing optimization* method effective in boosting logical reasoning performance?
- **RQ4**: How does DRI influence logical reasoning performance in reinforcement learning settings?

### 4.1 DATASETS AND SETTINGS

We conduct experiments on four logical reasoning benchmarks: *Reclor* (contextual reasoning) (Yu et al., 2020), *LogicBench* (deductive reasoning) (Parmar et al., 2024), *LogiQA* (workplace logic analysis) (Liu et al., 2020), and *LogiQA2.0* (enhanced adversarial patterns) (Liu et al., 2023). For our experiments, we utilize two backbone models: *LLaMA3.1-8B-Instruction* (Grattafiori et al., 2024) and *Qwen2.5-7B-Instruction* (Qwen et al., 2025). We refer to them hereafter as *LLaMA3.1-8B* and *Qwen2.5-7B*. Additionally, we include GPT-4 (OpenAI, 2023) and DeepSeek-V3 (DeepSeek-AI, 2024) as reference models. For more implementation details, please refer to the Appendix C.

## 4.2 Analysis of Data Reasoning Intensity (RQ1)

### 4.2.1 DRI Score Distribution

As shown in Figure 3(a), when the training set is binned into 20 score intervals, the sample counts form a bell-shaped curve with a weighted mean of $\mu = 0.526$ and standard deviation $\sigma = 0.204$. This approximately Gaussian frequency distribution demonstrates that our DRI scores effectively separate examples by their underlying reasoning potential. Figure 3(a) also plots model error rate against these intervals: error climbs from 8.5% at a score of 0.1 to 59.3% at 0.6, demonstrating a clear positive correlation between DRI and failure rate. Beyond a score of 0.6, error rate plateaus at approximately 61% ± 3.6%. This saturation likely arises because (1) very high DRI items exceed the model's current reasoning capacity, and (2) the top bins contain few, uniformly hard cases, so failures become uniformly pervasive. Unlike surface metrics (e.g., sentence length or vocabulary complexity), our score distribution captures deeper logical structure.

### 4.2.2 DRI Score Distribution Balancing

The original test set contains 4,743 examples. Its DRI score distribution (Figure 3(b)) reveals that 74.3% of samples fall into the mid-range interval `(0.2, 0.7)`, while only 17.2% lie in the high DRI interval `(>0.7)`. Such skew can introduce two evaluation biases: (1) Apparent overfitting in the mid-range: strong results in the overrepresented middle interval may conceal weaknesses on higher DRI examples; (2) Insufficient statistical power: the small number of high DRI samples ($n = 818$) yields wide confidence intervals for any observed gains. To correct this, we built a balanced benchmark (Figure 3(c)) by drawing 80 examples from each interval: two "extreme" bins (`<0.2`, `>0.9`) and contiguous 0.05-wide bins across `(0.2, 0.9)`. This uniform sampling ensures equal representation across the DRI spectrum, eliminating bias and enabling reliable comparisons.

### 4.2.3 Data Potential Analysis by DRI

Building on our validation of the DRI score and the creation of a balanced test benchmark, we next investigated how filtering training examples by their DRI scores affects learning. As shown in Figure 4, we systematically evaluate model performance when trained on subsets defined by DRI intervals, using both the original and balanced test sets. These experiments reveal three critical patterns, which can be summarized as the following observations:

- **Obs 1**: *Low DRI data can be safely pruned.* Both Figure 4(a) and (b) show that training on `Range(0.2, 1.0)`, which omits only the lowest 20% of examples, consistently outperforms full-data training, reducing error rates in nearly every bin. Even when further restricting to `Range(0.2, 0.8)` (removing both the lowest and highest 20%), overall accuracy remains on par with or slightly above the full-data baseline ($\Delta = +0.3\%$), and mid-range bins `(0.2, 0.7)` see notable gains. These results confirm that low DRI examples can be pruned without harming model performance and can often lead to improvements, while indiscriminate use of all data may introduce noise.
- **Obs 2**: *High DRI data are catalysts for improvement.* Figure 4(a) shows that training on `Range(0.2, 1.0)` yields the lowest error rates across all bins. When the upper bound is reduced to `Range(0.2, 0.9)`, error rates rise, particularly in the highest DRI bins. Narrowing further to `Range(0.2, 0.8)` causes a further increase in errors. This stepwise degradation, also reflected in Figure 4(b), confirms that examples with the highest DRI scores drive the most significant performance gains and act as catalysts for model learning.
- **Obs 3**: *Too little data breaks the learning process.* Figures 4(a) and (b) show that restricting training to narrow DRI intervals `Range(0.3, 0.7)` or `Range(0.4, 0.7)` leads to severe performance degradation, especially in the mid DRI range `(0.2, 0.7)`, where error rates exceed those from broader training ranges. This demonstrates that sparse coverage of the DRI spectrum impairs the model's ability to internalize and apply core reasoning patterns. The sharp drop in performance underscores the necessity of preserving sufficient data diversity and quantity across all DRI levels to sustain learning.

In summary, these results indicate that training data holds untapped potential and that model's logical reasoning performance is far from its ceiling. They also reveal limitations of static interval filtering: fixed cutoff thresholds can discard valuable examples or preserve noise. To address these issues,

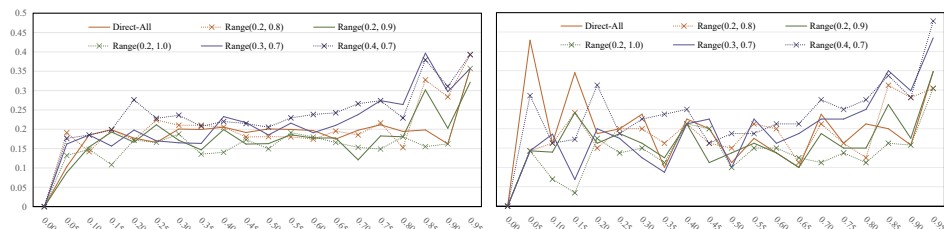

Figure 4: Experimental results of fine-tuning models in different intervals. *"Direct-All"* denotes a model fine-tuned on all training examples. *"Range(x, y)"* denotes a model fine-tuned on examples whose DRI scores fall between $x$ and $y$. Left: Testing set experiment results. Right: Balanced testing set experiment results. The horizontal axis represents score intervals, and the vertical axis represents error rates (lower error indicates better performance).

we propose the *re-cognizing optimization* framework, which first recalibrates the model's logical reasoning schema and then leverages DRI scores to reinforce its logical reasoning pathways, thereby unlocking the training data's potential and driving continuous improvements in the model's logical reasoning performance.

### 4.3 RE-COGNIZING OPTIMIZATION (RQ2)

Drawing on classic theories of human learning and resource allocation, we propose a two-phase *re-cognizing optimization* strategy guided by DRI scores. First, Sweller's cognitive load optimization principle (Sweller, 1988) suggests structuring learning so that basic schemas are established before tackling harder tasks, minimizing extraneous load. Second, the resource-rational analysis (Lieder & Griffiths, 2020) shows that humans allocate effort proportional to task demands, achieving an optimal balance of effort and reward. We leverage these insights as follows.

#### 4.3.1 PHASE I: MODEL COGNITION RESHAPING

In this phase, we reorder the training data according to DRI scores to "reset" and align the model's reasoning framework, applying Sweller's cognitive load theory. Allowing the model to explore the full spectrum of DRI examples from the outset helps it form broad reasoning patterns. This "low-stakes exploration" mirrors how human learners build foundational knowledge before tackling more challenging tasks, minimizing extraneous cognitive load and establishing a robust framework for subsequent learning.

#### 4.3.2 PHASE II: COGNITIVE REASONING ENHANCEMENT

Here, we implement resource-rational analysis by guiding the model's focus according to normalized DRI scores. The probability $p$ that the model attends to sample $i$ is calculated as follows:

$$p_i \ = \ \frac{\hat{s}_i}{\sum_{j=1}^{N} \hat{s}_j}, \quad \hat{s}_i = \frac{s_i - s_{\min}}{s_{\max} - s_{\min}}, \tag{9}$$

where $s_i$ is the raw DRI score of sample $i$, $s_{\min}$ and $s_{\max}$ are the minimum and maximum scores in the dataset, $\hat{s}_i$ is the normalized score for sample $i$, and the denominator $\sum_{j=1}^{N} \hat{s}_j$ sums these normalized scores over all $N$ samples. The *re-cognizing optimization* stratege echoes Lieder's insight that "human cognition allocates limited resources to maximize expected reasoning gains relative to reasoning demands". Through cognition-driven emphasis, the model is steered toward high DRI examples, thus securing greater learning returns under constrained resources.

### 4.4 RESULTS AND ANALYSIS (RQ3)

We evaluate the effectiveness of the *re-cognizing optimization* strategy by comparing it with static baselines and other data-centric methods across datasets. Other methods include curriculum learning (Bengio et al., 2009) and bin-based progressive learning (Klamkin et al., 2024). In curriculum learning, training examples are introduced in order of increasing DRI scores; in bin-based progressive

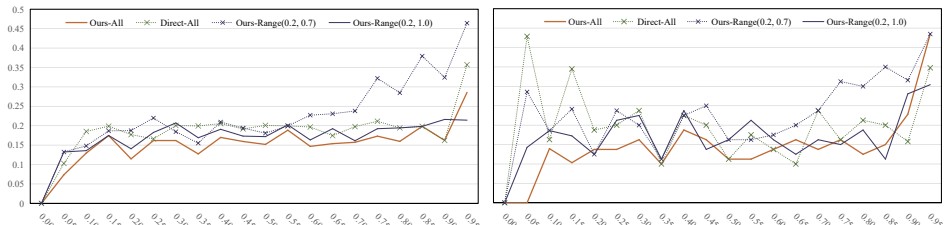

Figure 5: Experimental results of fine-tuning models using different methods. *"Ours-All"*: model fine-tuned on the full training set using *re-cognizing optimization*. *"Ours-Range(x, y)"*: model fine-tuned with *re-cognizing optimization* on examples whose DRI scores fall between $x$ and $y$. Left: Test dataset results. Right: Balanced testset results. The horizontal axis represents score intervals and the vertical axis represents error rates (lower error indicates better performance).

| Model | Methods | Unbalanced | | | | | Balanced | | | | |
|---|---|---|---|---|---|---|---|---|---|---|---|
| | | Recolr | LogiQA | LogiQA2.0 | LogicBench | Avg. | Recolr | LogiQA | LogiQA2.0 | LogicBench | Avg. |
| Closed-source LLMs | GPT-4 | 0.808 | 0.525 | 0.664 | 0.774 | 0.707 | 0.794 | 0.588 | 0.651 | 0.765 | 0.699 |
| | DeepSeek-V3 | 0.754 | 0.484 | 0.663 | 0.814 | 0.712 | 0.738 | 0.500 | 0.663 | 0.847 | 0.686 |
| LLaMA3.1-8B | Base | 0.444 | 0.347 | 0.378 | 0.707 | 0.521 | 0.447 | 0.412 | 0.383 | 0.714 | 0.490 |
| | Directly | 0.896 | 0.724 | 0.805 | 0.812 | 0.806 | 0.901 | 0.774 | 0.802 | 0.812 | 0.816 |
| | Curriculum Learning | 0.926 | 0.634 | 0.798 | 0.854 | 0.811 | 0.908 | 0.684 | 0.773 | 0.864 | 0.815 |
| | Bin-based Progressive Learning | 0.930 | 0.742 | 0.821 | 0.831 | 0.826 | 0.894 | 0.763 | 0.809 | 0.836 | 0.823 |
| | Re-Cognizing Optimization (Ours) | 0.930 | **0.750** | **0.835** | **0.857** | **0.843** | **0.922** | **0.775** | **0.840** | **0.865** | **0.851** |
| | Re-Cognizing Optimization *w/o* Stage1 | 0.904 | 0.699 | 0.791 | 0.747 | 0.771 | 0.918 | 0.726 | 0.825 | 0.763 | 0.806 |
| | Re-Cognizing Optimization *w/o* Stage2 | 0.910 | 0.717 | 0.796 | 0.824 | 0.809 | 0.918 | 0.768 | 0.834 | 0.859 | 0.844 |
| Qwen2.5-7B | Base | 0.466 | 0.372 | 0.424 | 0.629 | 0.509 | 0.433 | 0.435 | 0.409 | 0.678 | 0.490 |
| | Directly | 0.786 | 0.743 | 0.748 | 0.810 | 0.778 | 0.797 | 0.719 | 0.788 | 0.888 | 0.798 |
| | Curriculum Learning | 0.864 | 0.750 | 0.806 | 0.850 | 0.823 | 0.872 | 0.739 | 0.797 | 0.919 | 0.831 |
| | Bin-based Progressive Learning | 0.890 | 0.730 | 0.782 | 0.842 | 0.812 | 0.911 | 0.697 | 0.806 | 0.719 | 0.833 |
| | Re-Cognizing Optimization (Ours) | **0.944** | **0.824** | **0.845** | **0.858** | **0.858** | **0.948** | **0.797** | **0.884** | **0.919** | **0.886** |
| | Re-Cognizing Optimization *w/o* Stage1 | 0.856 | 0.808 | 0.817 | 0.833 | 0.827 | 0.833 | 0.790 | 0.822 | 0.894 | 0.835 |
| | Re-Cognizing Optimization *w/o* Stage2 | 0.798 | 0.679 | 0.767 | 0.827 | 0.784 | 0.839 | 0.684 | 0.778 | 0.872 | 0.793 |

Table 1: Experimental results from different test sets. Our *Re-Cognizing Optimization* method is compared against other approaches on both LLaMA3.1-8B and Qwen2.5-7B using accuracy as the evaluation metric. Results are reported on both unbalanced and balanced test sets, with the best performance in each setting highlighted in bold.

learning, data is divided into different DRI bins, and the next DRI bin is introduced only after the model has been trained on the current bin. Table 1 summarizes the performance across datasets, while Figure 5 shows the error rates across DRI bins for the original and balanced test sets. Figure 6 (left) compares our method with the other two methods. From the experimental results, we have the following observations:

- **Obs 4:** *Re-cognizing optimization consistently reduces errors across datasets and DRI bins, leading to comprehensive and robust performance gains.* In Figure 5, Figure 6 (left), and Table 1, our method consistently achieves the best performance across datasets and test splits, confirming its ability to generalize dataset-specific reasoning patterns. For example, in the lowest DRI interval (0, 0.2), errors drop from 17.6%/25% to 12.6%/11.2% on the original/balanced test sets, demonstrating reduced overfitting to trivial cases. This trend holds across all DRI bins, resulting in a systematic reduction in errors across the spectrum. On datasets like LogiQA and LogiQA2.0, where baseline accuracies are lower, our method shows a significant improvement over others, demonstrating its adaptability and robustness on diverse datasets.
- **Obs 5:** *Re-cognizing optimization demonstrates resilience in data-scarce settings by maintaining strong performance even with restricted training ranges.* In Figure 5, when we apply *re-cognizing optimization* to restricted training sets—Our-Range(0.2, 0.7) and Our-Range(0.2, 1.0)—we observe a 13.6% error increase when high DRI examples (>0.7) are omitted. However, our method effectively compensates for this loss, maintaining performance comparable to full-data training within the Our-Range(0.2, 0.7) subset ($\Delta = +2.5\%$). Expanding the training range to include higher DRI examples (Our-Range(0.2, 1.0)) consistently maintains lower error rates than direct full-data training, validating the predictive power of our DRI and demonstrating that our method can effectively improve performance even in data-scarce settings.

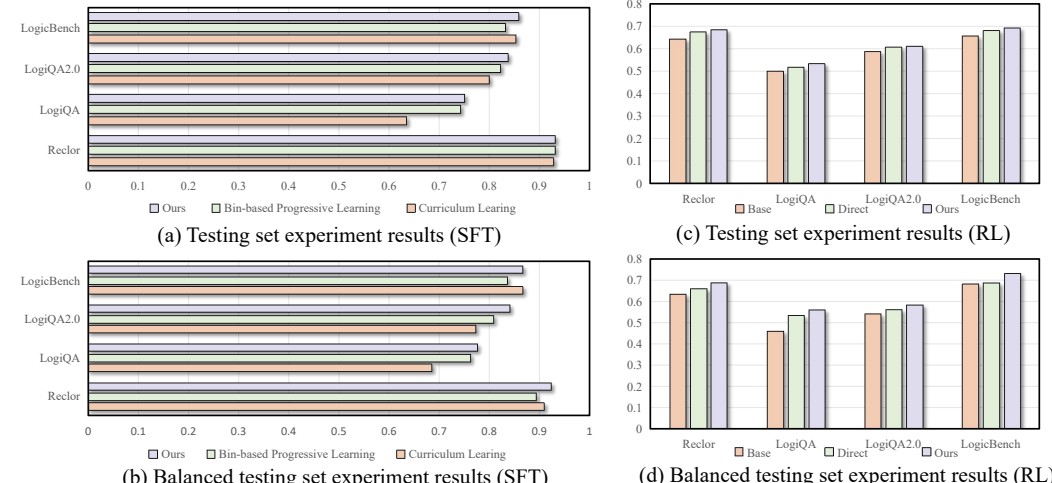

(a) Testing set experiment results (SFT)

(c) Testing set experiment results (RL)

(b) Balanced testing set experiment results (SFT)

(d) Balanced testing set experiment results (RL)

Figure 6: Experimental results of SFT and RL across different dataset. Accuracy is used as the evaluation metric (higher is better). Both SFT and RL methods are evaluated on the original and balanced test sets. (a) SFT on the original test set: Ours vs. Bin-based Progressive Learning vs. Curriculum Learning. (b) SFT on the balanced test set: same comparisons. (c) RL on the original test set: Ours vs. Base vs. Direct static reward). (d) RL on the balanced test set: same comparisons. In every setting, our method (Ours) achieves the highest accuracy, demonstrating its superiority under both SFT and RL regimes.

**Ablation Study** To assess the contribution of each training stage, we conduct ablation experiments by removing either Stage 1 or Stage 2 from the full *re-cognizing optimization* pipeline. As shown in Table 1, both variants lead to substantial performance drops on both LLaMA3.1-8B and Qwen2.5-7B. This confirms that both stages are essential for achieving optimal reasoning performance. In summary, our results show our *re-cognizing optimization* effectively directs the model to high DRI examples, boosting learning efficiency. It systematically reduces errors across DRI bins (proving improved reasoning) and outperforms traditional methods like curriculum and bin-based learning, demonstrating flexibility and robustness across scenarios.

### 4.5 EXPANSION EXPLORATION (RQ4)

Some studies (Chen et al., 2025a; Chu et al., 2025; Chen et al., 2025b) suggest that supervised fine-tuning (SFT) and reinforcement learning (RL) play different roles in stimulating the capabilities of language models. To assess the impact of our DRI scores in an RL setting, we adopt the GRPO algorithm (Shao et al., 2024) within the TRL framework (von Werra et al., 2020) and apply LoRA-based parameter-efficient tuning (Hu et al., 2022). We provide proportionally larger accuracy rewards for samples with higher DRI scores, incentivizing the model to handle more reasoning-intensive cases and internalize richer reasoning patterns. We also enforce a structured output format and reward the model for both format compliance and the quality of its reasoning trace. Figure 6(c) and (d) compare our method against two baselines: the base model without RL and a direct variant with fixed accuracy rewards. In both the original and balanced test sets, our DRI-guided method consistently outperforms these baselines, confirming that dynamic, score-based rewards combined with structured output incentives significantly enhance the model's reasoning capabilities.

## 5 CONCLUSION

This work introduces a data-centric framework for enhancing LLM reasoning by proposing a novel metric—*Data Reasoning Intensity* (DRI)—to quantify the inherent logical complexity within training samples. We further present the *re-cognizing optimization* strategy, optimizing training data to better align with LLMs' reasoning boundaries. Our framework is validated across multiple reasoning benchmarks, demonstrating consistent improvements over existing methods. Additionally, we show that enhancing DRI benefits both supervised and reinforcement learning settings. We hope this study offers new insights into measuring and activating the untapped reasoning potential of LLMs, and inspires future work on cognitive-level data optimization.

# 6 ETHICS STATEMENT

Our work investigates the role of *data reasoning intensity* in shaping the reasoning capabilities of LLMs. While the goal of this research is to promote a deeper understanding of data-centric optimization and enable more reliable reasoning behaviors, we acknowledge potential risks if the techniques are misapplied, such as amplifying biased reasoning patterns or reinforcing spurious correlations in data. We emphasize that our approach is intended to improve transparency and effectiveness of training rather than to manipulate or distort reasoning outcomes. Researchers applying our method should adopt responsible practices, including careful data curation and validation, to avoid negative societal impacts.

# 7 REPRODUCIBILITY STATEMENT

To ensure the reproducibility of our findings, we provide detailed implementation instructions in Appendix C. The complete source code is included in the supplementary file, enabling other researchers to replicate and verify our experiments. Furthermore, we describe the use of LLMs in Appendix A to maintain transparency in our methodology. These measures are intended to facilitate rigorous validation and encourage further research building upon our work.

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

# Appendix

## Table of Contents

## A   THE USE OF LARGE LANGUAGE MODELS

In accordance with the ICLR 2026 policies on the use of LLMs, we disclose that LLMs were employed solely for translation and language refinement purposes. All research ideas, experimental design, implementation, analysis, and conclusions are the sole responsibility of the authors. We have carefully verified the accuracy and integrity of the manuscript to ensure that no false or misleading content was introduced by the use of LLMs.

## B   LIMITATION, BROADER IMPACT, AND FUTURE WORK

We acknowledge that although our *data reasoning intensity* score and *re-cognizing optimization* method are effective for several tasks in our research, they are far from being perfect. Here, we honestly discuss the limitations, broader impact, and potential avenues for future works.

### B.1   LIMITATION

One limitation of our work is the relatively small scale and variety of models we evaluated—due to budget and time constraints, we focused on only two model families (e.g., LLaMA3.1-8B and Qwen2.5-7B), which may limit the generalizability of our findings to larger or more diverse architectures. Additionally, while our *data reasoning intensity* score provides a useful proxy for reasoning potential, its formulation could be refined further: the current metrics may not capture all aspects of reasoning complexity or transfer seamlessly to other task domains. Moreover, our logical element extraction relies on LLM-based distillation functions, which can introduce noise or inaccuracies; nevertheless, we found these errors to be minor and within acceptable bounds, having minimal impact on overall metric reliability. Finally, our use of GRPO-based reinforcement learning was exploratory and preliminary; more extensive experiments with alternative reward schemes, longer training runs, and varied model capacities will be necessary to fully assess the robustness and scalability of *re-cognizing*

*optimization*. We also did not explore individualized learning trajectories emphasized in cognitive science, nor use our reasoning-intensity signal to dynamically switch between System 1 and System 2 modes. Moreover, we restrict our evaluation to fine-tuning existing pretrained checkpoints rather than full-scale pretraining, since retraining multi-billion-parameter models from scratch requires resources beyond our current capabilities.

## B.2 BROADER IMPACT

Our *data reasoning intensity* score is a pioneering metric that quantifies each example's reasoning potential. Paired with our *re-cognizing optimization* framework, which uses this score to guide training, our approach reduces unnecessary computation, boosts model reasoning performance, and supports more sustainable AI practices. We believe this work will inspire new directions and offer systematic guidance for future research on unlocking the latent potential of LLM training data. Societally, our method can positively influence the efficient training of LLMs, enabling the creation of more robust and resource-efficient AI systems.

## B.3 FUTURE WORK

**Broader model and task coverage.** In future work, it would be valuable to evaluate our framework on additional architectures—beyond LLaMA-7B and Qwen2.5-7B—and across new domains such as mathematical reasoning, code generation, and multimodal understanding. This broader testing would help establish the generality and limits of *data reasoning intensity* and *re-cognizing optimization*.

**Refined reinforcement learning integration.** While we have already incorporated reasoning-intensity scores into GRPO rewards, it would be useful to explore more sophisticated applications—such as dynamic reward shaping, alternative RL algorithms, or multi-objective formulations—to further boost learning efficiency and stability.

**Adaptive reasoning routing.** We also plan to investigate using reasoning-intensity as a runtime signal to guide the model's choice of reasoning mode, enabling dynamic switching between fast, heuristic processing (System 1) and deeper, deliberative reasoning (System 2). This may prevent overthinking on trivial inputs and ensure adequate effort on challenging ones.

**Compatibility with alternative data-selection methods.** Our *data reasoning intensity* metric and *re-cognizing optimization* are fully compatible with other sampling strategies. In future work, we will experiment with hybrid schemes that combine our method with these complementary approaches to maximize sample efficiency and model performance and to provide a more rigorous comparison against established data-selection techniques.

**Pretraining-stage integration.** An exciting direction is to extend reasoning-intensity guidance into the pretraining curriculum. Although pretraining multi-billion-parameter models from scratch exceeds our current resources, future work could integrate *data reasoning intensity* into early model training to shape core reasoning capabilities, contingent on access to sufficient compute.

## C DETAILS FOR THE MAIN EXPERIMENTS

### C.1 TRAINING CONFIGURATION

Our datasets consist of 36,788 training instances and 4,743 test instances, covering a broad range of logical challenges, including propositional logic, syllogisms, and temporal reasoning.

Both *LLaMA3.1-8B* and *Qwen2.5-7B* are fine-tuned using supervised fine-tuning (SFT) under the TRL framework, employing LoRA modules with a rank of 64, $\alpha = 16$, and a dropout rate of 0.05. In addition to the SFT experiments, we also conduct reinforcement learning (RL) experiments on *Qwen2.5-7B*. For RL, we apply the GRPO algorithm under the TRL framework, with LoRA configured as rank 16, $\alpha = 32$, and dropout rate 0.05. Due to resource constraints, we used a random subset of 1,000 training samples for the training. All evaluations across both settings are performed in a zero-shot manner.

For both direct training and *re-cognizing optimization*, we use the Paged AdamW optimizer with a learning rate of 2e-4, $\beta_1 = 0.9$, $\beta_2 = 0.95$, perform gradient clipping at 0.3 with a warmup ratio

of 10%, set the global batch size to 16 (8 per device × 2 accumulation steps), and conduct bfloat16 mixed - precision training.

For GRPO experiments, we use the AdamW optimizer with a learning rate of 5e-5, perform gradient clipping at 0.3, apply a 15% warmup over 100 steps, set weight decay to 0.01, use a global batch size of 8 (2 per device × 4 accumulation steps), enable fp16 mixed-precision training, employ a cosine-with-restarts learning-rate schedule, generate 4 completions per prompt with a maximum length of 512 tokens at temperature 0.9, and scale rewards.

**Compute Resources**   Our *re-cognizing optimization* experiments were run on eight NVIDIA RTX 4090d GPUs (though a single 4090d can support smaller runs), with each full training run taking approximately 6 hours. GRPO-based RL experiments used four NVIDIA A800 GPUs (minimum one A800 required) and averaged around 10 hours per run.

## C.2   EXAMPLES OF DIFFERENT SCORE DATA

---

**Examples of Different Score Data**

"context": "john knows how to play the piano"
"question": "does this entail that someone has the ability to play the piano?"
"answer": "yes"
"ReasoningIntensityScore": 0.05729995978985077

---

"context": "Roves had held a senior position in the Navy before taking office.One of his good friends asked him about the Navy's plan to establish a submarine base on an island.Roosevelt looked around mysteriously and asked in a low voice.Čan you keep it a secret?" "Of course I can!" The friend was very sure."So," Roosevelt said with a smile, "I can too.""
"question": "This text tells us:"
"options": ["Detours can also achieve the goal."
"Humor can subtly solve problems"
"Adherence to principles and flexibility are not contradictory."
"Don't do anything to others"],
"answer": "1"
"ReasoningIntensityScore": 0.589367067922439

---

"context": "It is necessary to pay attention to avoiding hollowing out in the development of the service industry, but it is wrong and dangerous to think that the rapid development of modern service industry in China's economic growth will definitely lead to a hollowing out of the industry.This view of China will make China's economy lose an important window period for the rapid development of the modern service industry.In fact, the formation of an industrial structure dominated by the service industry does not mean the decline of the status of the manufacturing industry, nor does it mean "de-industrialization" "It is not the same as starting the hollowing out process of the industry."
"question": "The main emphasis of this text?"
"options": ["The rapid development of modern service industry cannot lead to a hollowing out of the industry"
"How to objectively evaluate the advantages and disadvantages of the rapid development of modern service industry"
"Whether it will cause industrial hollowing depends on the prosperity of the manufacturing industry"
"Don't worry about the hollowness of the industry and miss the opportunity to develop the service industry"],
"answer": "3"
"ReasoningIntensityScore": 0.9464210784935705

## C.3 REASONING INTENSITY SCORE CALCULATION PROMPT

> **Prompt for Logical Decomposition**
>
> Instructions: Please extract the predicates and constants from the following context and create logical expressions that represent the relationships described. Format the output as a dictionary where each entry is a list of items. Follow these specific rules:
> 1. **Extract predicates** as core action words or relationships defining connections between entities, and output them as a list under the 'Predicates' key.
> 2. **Extract constants** as the specific entities or values mentioned in the context, and output them as a list under the 'Constants' key.
> 3. **Create logical expressions** using the extracted predicates and constants. Each logical expression should be simple and based on a single predicate. Output them as a list under the 'Logical Expressions' key.
> 4. Ensure that the output strictly follows the format provided in the example.
> 5. Do not combine expressions using logical operators such as 'and,' 'or,' etc., unless the relationship is explicitly mentioned in the context.
> Example:
> [Context: If an individual consumes a significant amount of water, they will experience a state of hydration. Conversely, if excessive amounts of sugar are ingested, a sugar crash will ensue. It is known that at least one of the following statements is true: either Jane consumes ample water or she will not experience a sugar crash. However, the actual veracity of either statement remains ambiguous, as it could be the case that only the first statement is true, only the second statement is true, or both statements are true.]
> [Output:
> Predicates: ['Consumes(x, y)': Represents the act of 'x' consuming 'y' (e.g., an individual consuming water or sugar).,
> 'ExperienceState(x, y)': Represents 'x' experiencing a state 'y' (e.g., hydration).,
> 'Ingested(x, y)': Represents 'x' ingesting 'y' (e.g., excessive sugar).,
> 'Ensue(x)': Represents that a condition 'x' follows or results (e.g., a sugar crash).,
> 'TrueStatement(x)': Indicates that 'x' is known to be true.,
> 'NotExperience(x, y)': Represents 'x' not experiencing a condition 'y' (e.g., not experiencing a sugar crash).],
> Constants: ['Individual': Represents a generic person in the context.,
> 'Water': The substance being consumed by an individual.,
> 'Hydration': The state that results from sufficient water consumption.,
> 'Sugar': A substance that can be ingested.,
> 'SugarCrash': The state that follows excessive sugar intake.,
> 'Jane': A specific person mentioned in the context.],
> Logical Expressions: [Consumes(Individual, Water)',
> 'ExperienceState(Individual, Hydration)',
> 'Ingested(Individual, Sugar)',
> 'Ensue(SugarCrash)',
> 'Consumes(Jane, Water)',
> 'NotExperience(Jane, SugarCrash)',
> 'TrueStatement(Consumes(Jane, Water) $\lor \neg$Experience(Jane, SugarCrash))'
> —
> ** Tips for Extracting Predicates, Constants, and Logical Expressions:
> - Focus on identifying core action or relational words for predicates.
> - Extract constants as the specific entities mentioned.
> - Use variables ('x', 'y', etc.) to generalize when needed.
> - Ensure logical expressions are complete and accurately reflect relationships.
> ** Your Task:
> Context: [context]

## Prompt for Combinatorial Reasoning in BQA

Instructions: Please analyze the following binary question data. Since BQA is treated as an MCQA with a single option derived from the question, perform the analysis for this single option. Extract the relevant preconditions, define the deduction target, outline the deduction steps based on the provided Predicates, Constants, and Logical Expressions, and determine whether the option is correct based on the given answer index. Follow these specific rules:

1. **Output** should be a JSON object containing only the 'option_analysis' field, which is a list of analyses for each option.

2. For the option_analysis, include the following fields:

- option_index: The index of the option (0-based).

- option_text: The text of the option (extracted from the question).

- preconditions: A list of relevant preconditions from the Logical Expressions that pertain to the option.

- deduction_target: The abstracted logical conclusion that the option is attempting to establish.

- deduction_steps: A step-by-step logical deduction process from the preconditions to the deduction target. Each step should include:

- step: The step number.

- task: A description of what is being checked or inferred in this step.

- expression: The logical expression used in this step, enclosed in backticks.

- result: The outcome of this step, enclosed in backticks.

- If a deduction step cannot proceed due to unsupported premises, indicate the failure and terminate further steps.

- is_correct: A boolean indicating whether the option is correct (true) or not (false). This should align with the answer field. When 'answer' is equal to 'yes', the value is true, otherwise the value is false.

3. **Format Requirements**:

- The output must strictly follow the JSON structure as shown in the example.  - Ensure consistency in field naming and hierarchy.

- Do not include any additional fields or information not specified in the example.

4. **Important Considerations**:

- Only include preconditions that are directly relevant to the option being analyzed.

- Maintain logical rigor in deduction steps, ensuring each step follows from the previous ones based on the preconditions.  - Avoid including unrelated preconditions to minimize complexity and enhance clarity.

Example:

- context: All people who regularly drink coffee are dependent on caffeine. People either regularly drink coffee or joke about being addicted to caffeine. No one who jokes about being addicted to caffeine is unaware that caffeine is a drug. Rina is either a student and unaware that caffeine is a drug, or neither a student nor unaware that caffeine is a drug. If Rina is not a person dependent on caffeine and a student, then Rina is either a person dependent on caffeine and a student, or neither a person dependent on caffeine nor a student.

- question: Rina is a person who jokes about being addicted to caffeine or unaware that caffeine is a drug.

- options: Rina is a person who jokes about being addicted to caffeine or unaware that caffeine is a drug.

- answer: yes

- Predicates:

- RegularlyDrink(x, y): Represents 'x' regularly drinking 'y' (e.g., a person regularly drinking coffee).

- DependentOn(x, y): Represents 'x' being dependent on 'y' (e.g., a person dependent on caffeine).

- JokeAbout(x, y): Represents 'x' joking about 'y' (e.g., a person joking about being addicted to caffeine).

- UnawareThat(x, y): Represents 'x' being unaware that 'y' (e.g., a person unaware that caffeine is a drug).

- IsStudent(x): Represents 'x' being a student.

- IsNeither(x, y): Represents 'x' being neither 'y' (used for expressing negation of multiple conditions).
- Constants:
- People: Generic individuals.
- Coffee: The beverage being consumed.
- Caffeine: The substance people can be dependent on.
- Rina: A specific person mentioned in the context.
- Logical Expressions:
- $\text{DependentOn}(\text{People}, \text{Caffeine}) \Rightarrow \text{RegularlyDrink}(\text{People}, \text{Coffee})$
- $\text{JokeAbout}(\text{People}, \text{Caffeine}) \lor \text{RegularlyDrink}(\text{People}, \text{Coffee})$
- $\text{UnawareThat}(\text{People}, \text{Caffeine}) \Rightarrow \text{JokeAbout}(\text{People}, \text{Caffeine})$
- $\text{IsStudent}(\text{Rina}) \land \text{UnawareThat}(\text{Rina}, \text{Caffeine}) \lor \neg\text{IsStudent}(\text{Rina}) \land \neg\text{UnawareThat}(\text{Rina}, \text{Caffeine})$
- $\neg\text{DependentOn}(\text{Rina}, \text{Caffeine}) \land \text{IsStudent}(\text{Rina}) \Rightarrow (\text{DependentOn}(\text{Rina}, \text{Caffeine}) \land \text{IsStudent}(\text{Rina})) \lor \neg(\text{DependentOn}(\text{Rina}, \text{Caffeine}) \land \text{IsStudent}(\text{Rina}))$

Output:
option_analysis:
- **Option Index**: 0
- **Option Text**: Rina is a person who jokes about being addicted to caffeine or unaware that caffeine is a drug.
- **Preconditions**:
- $\text{JokeAbout}(\text{People}, \text{Caffeine}) \lor \text{RegularlyDrink}(\text{People}, \text{Coffee})$
- $\text{UnawareThat}(\text{People}, \text{Caffeine}) \Rightarrow \text{JokeAbout}(\text{People}, \text{Caffeine})$
- **Deduction Target**: $\text{JokeAbout}(\text{Rina}, \text{Caffeine}) \lor \text{UnawareThat}(\text{Rina}, \text{Caffeine})$
- **Deduction Steps**:
1. **Step**: 1
- **Task**: Instantiate the general disjunction for Rina from the population-level statement.
- **Expression**: $\text{JokeAbout}(\text{Rina}, \text{Caffeine}) \lor \text{RegularlyDrink}(\text{Rina}, \text{Coffee})$
- **Result**: Derived from $\text{JokeAbout}(\text{People}, \text{Caffeine}) \lor \text{RegularlyDrink}(\text{People}, \text{Coffee})$
2. **Step**: 2
- **Task**: Apply the implication that joking about caffeine addiction leads to being unaware that caffeine is a drug for Rina.
- **Expression**: $\text{UnawareThat}(\text{Rina}, \text{Caffeine}) \Rightarrow \text{JokeAbout}(\text{Rina}, \text{Caffeine})$
- **Result**: If Rina jokes about caffeine, then Rina is unaware that caffeine is a drug.
3. **Step**: 3
- **Task**: Combine the instantiated disjunction with the implication to derive the final conclusion.
- **Expression**: $\text{JokeAbout}(\text{Rina}, \text{Caffeine}) \lor \text{UnawareThat}(\text{Rina}, \text{Caffeine})$
- **Result**: Since $\text{JokeAbout}(\text{Rina}, \text{Caffeine})$ implies $\text{UnawareThat}(\text{Rina}, \text{Caffeine})$, the disjunction holds.
- **Is Correct**: True

Tips for Option Analysis:
- **Preconditions**: Only include logical expressions that are directly relevant to the option being analyzed. Avoid listing all possible preconditions.
- **Deduction Steps**: Ensure each step logically follows from the previous one based on the preconditions. If a step cannot be completed due to insufficient support from the preconditions, indicate the failure and stop further deductions for that option.
- **is_correct**: This field should be true only for the option that matches the answer field. Since BQA has only one option, is_correct should align with the answer field. - **Format Consistency**: Maintain the same JSON structure and field naming conventions across all options to ensure uniformity and ease of data extraction.
- **Logical Accuracy**: Ensure that all logical expressions and deductions accurately reflect the relationships defined by the predicates and constants.

Your Task:
Analyze the following Input data and generate the option_analysis section as per the example above. Replace the xxx placeholders in the example with actual data derived from the input.
Input Data:

input_data_here
Please generate the option_analysis section based on the above input data.

## Prompt for Combinatorial Reasoning in MCQA

Instructions: Please analyze the following multiple-choice question data. For each option, extract the relevant preconditions, define the deduction target, outline the deduction steps based on the provided Predicates, Constants, and Logical Expressions, and determine whether the option is correct according to the given answer index. Follow these specific rules:

1. The **Output** should be a JSON object containing only the 'option_analysis' field, which is a list of analyses for each option.

2. For each option in 'option_analysis', include the following fields:
- **option_index**: The index of the option (starting from 0).
- **option_text**: The text content of the option.
- **preconditions**: A list of relevant preconditions from the Logical Expressions that pertain to the option.
- **deduction_target**: The abstracted logical conclusion that the option is attempting to establish.
- **deduction_steps**: A step-by-step logical deduction process from the preconditions to the deduction target. Each step should include:
- **step**: The step number.
- **task**: A description of what is being checked or inferred in this step.
- **expression**: The logical expression used in this step, enclosed in backticks. Here, logical symbols like "implies" is represented as "$\Rightarrow$", "and" as "$\wedge$", "or" as "$\vee$", "not" as "$\neg$", "for all" as "$\forall$", "there exists" as "$\exists$" in LaTeX notation.
- **result**: The outcome of this step, enclosed in backticks.
- If a deduction step cannot proceed due to unsupported premises, indicate the failure and terminate further steps for that option.
- **is_correct**: A boolean indicating whether the option is correct (true) or not (false). The values of 'Answer' are '0', '1', '2', '3', where '0' represents the first option, '1' represents the second option, and so on. If the value of 'option_index' is the same as the value of 'Answer', then the value of 'is_correct' is 'true'.

3. **Format Requirements**:
- The output must strictly follow the JSON structure as shown in the example. - Ensure consistency in field naming and hierarchy.
- Do not include any additional fields or information not specified in the example.

4. **Important Considerations**:
- Only include preconditions that are directly relevant to the option being analyzed.
- Maintain logical rigor in deduction steps, ensuring each step follows from the previous ones based on the preconditions.
- Avoid including unrelated preconditions to minimize complexity and enhance clarity.

Example:
- **Context**: In rheumatoid arthritis, the body's immune system misfunctions by attacking healthy cells in the joints causing the release of a hormone that in turn causes pain and swelling. This hormone is normally activated only in reaction to injury or infection. A new arthritis medication will contain a protein that inhibits the functioning of the hormone that causes pain and swelling in the joints. - **Question**: The statements above, if true, most strongly support which one of the following conclusions?
- **Options**: 1. Unlike aspirin and other medications that reduce pain and swelling and that are currently available, the new medication would repair existing cell damage that had been caused by rheumatoid arthritis.
2. A patient treated with the new medication for rheumatoid arthritis could sustain a joint injury without becoming aware of it.
3. Joint diseases other than rheumatoid arthritis would not be affected by the new medication.
4. The benefits to rheumatoid arthritis sufferers of the new medication would outweigh the medication's possible harmful side effects.
- **Answer**: 1

- **Predicates**: - Attack$(x, y)$: Represents 'x' attacking 'y' (e.g., the immune system attacking healthy cells).
- Release$(x, y)$: Represents 'x' releasing 'y' (e.g., the release of a hormone).
- Cause$(x, y)$: Represents 'x' causing 'y' (e.g., the hormone causing pain and swelling).
- Activate$(x, y)$: Represents 'x' activating 'y' (e.g., the hormone being activated by injury or infection).
- Inhibit$(x, y)$: Represents 'x' inhibiting 'y' (e.g., the protein inhibiting the hormone).
- Contain$(x, y)$: Represents 'x' containing 'y' (e.g., the medication containing a protein).
- **Constants**: - ImmuneSystem: The body's defense mechanism.
- HealthyCells: Cells in the joints that are not diseased.
- Hormone: A chemical messenger involved in causing pain and swelling.
- Pain: A sensation caused by the hormone.
- Swelling: A condition caused by the hormone.
- Injury: A condition that normally activates the hormone.
- Infection: A condition that normally activates the hormone.
- ArthritisMedication: A new medication for treating arthritis.
- Protein: A component of the medication that inhibits the hormone.
- **Logical Expressions**: - Attack(ImmuneSystem, HealthyCells)
- Release(ImmuneSystem, Hormone)
- Cause(Hormone, Pain)
- Cause(Hormone, Swelling)
- Activate(Injury, Hormone)
- Activate(Infection, Hormone)
- Inhibit(Protein, Hormone)
- Contain(ArthritisMedication, Protein)
Option Analysis
1. **Option Index**: 0
- **Option Text**: Unlike aspirin and other medications that reduce pain and swelling and that are currently available, the new medication would repair existing cell damage that had been caused by rheumatoid arthritis.
- **Preconditions**:
- Attack(ImmuneSystem, HealthyCells)
- Release(ImmuneSystem, Hormone)
- Cause(Hormone, Pain)
- Cause(Hormone, Swelling)
- **Deduction Target**: Repair(ArthritisMedication, HealthyCellsDamage)
- **Deduction Steps**:
- **Step 1**:
- **Task**: Check if Attack(ImmuneSystem, HealthyCells) implies Damage(HealthyCells).
- **Expression**: Attack(ImmuneSystem, HealthyCells) $\Rightarrow$ Damage(HealthyCells)
- **Result**: Supported by context (immune system attacking healthy cells causes damage).
- **Step 2**:
- **Task**: Check if Contain(ArthritisMedication, Protein) and Inhibit(Protein, Hormone) imply Repair(ArthritisMedication, HealthyCellsDamage).
- **Expression**: Contain(ArthritisMedication, Protein) $\wedge$ Inhibit(Protein, Hormone) $\Rightarrow$ Repair(ArthritisMedication, HealthyCellsDamage)
- **Result**: Not supported. The context only states that the protein inhibits the hormone, not that it repairs damage.
- **Step 3**:
- **Task**: Derivation fails.
- **Expression**: Derivation cannot proceed.
- **Result**: Repair(ArthritisMedication, HealthyCellsDamage) cannot be derived from the given preconditions.
- **Is Correct**: False
2. **Option Index**: 1
- **Option Text**: A patient treated with the new medication for rheumatoid arthritis could sustain a joint injury without becoming aware of it.

- **Preconditions**:
- Cause(Hormone, Pain)
- Activate(Injury, Hormone)
- Inhibit(Protein, Hormone)
- Contain(ArthritisMedication, Protein)
- **Deduction Target**: $\exists$ Patient, Injury : [Sustain(Patient, Injury) $\wedge$ Unaware(Patient, Injury)]
- **Deduction Steps**:
- **Step 1**:
- **Task**: Determine the effect of Inhibit(Protein, Hormone) from the medication.
- **Expression**: Inhibit(Protein, Hormone)
- **Result**: Supported by context: The protein inhibits the hormone that causes pain and swelling.
- **Step 2**:
- **Task**: Analyze the implication of inhibiting the hormone on pain and swelling.
- **Expression**: Inhibit(Protein, Hormone) $\Rightarrow$ $\neg$Cause(Hormone, Pain) $\wedge$ $\neg$Cause(Hormone, Swelling)
- **Result**: Supported by context: If the hormone is inhibited, it cannot cause pain and swelling.
- **Step 3**:
- **Task**: Infer the patient's awareness of injury when pain and swelling are absent.
- **Expression**: $\neg$Cause(Hormone, Pain) $\wedge$ $\neg$Cause(Hormone, Swelling) $\Rightarrow$ Unaware(Patient, Injury)
- **Result**: Supported by context: Without pain and swelling, the patient may not be aware of sustaining an injury.
- **Step 4**:
- **Task**: Combine the above implications to conclude the deduction target.
- **Expression**: $\exists$ Patient, Injury : [Sustain(Patient, Injury) $\wedge$ Unaware(Patient, Injury)] -
- **Result**: Deduction is valid based on the inhibited hormone preventing awareness of injury.
- **Is Correct**: True
3. **Option Index**: 2
- **Option Text**: Joint diseases other than rheumatoid arthritis would not be affected by the new medication.
- **Preconditions**:
- Inhibit(Protein, Hormone)
- Contain(ArthritisMedication, Protein)
- **Deduction Target**: $\forall x$[JointDisease($x$) $\wedge$ $x$ $\neq$ RheumatoidArthritis $\Rightarrow$ $\neg$Affect(ArthritisMedication, $x$)]
- **Deduction Steps**:
- **Step 1**:
- **Task**: Check if the context provides information about other joint diseases.
- **Expression**: JointDisease($x$) $\wedge$ $x \neq$ RheumatoidArthritis
- **Result**: Not supported. The context only discusses rheumatoid arthritis.
- **Step 2**:
- **Task**: Determine if there is any implication that the medication specifically targets rheumatoid arthritis.
- **Expression**: Contain(ArthritisMedication, Protein) $\Rightarrow$ SpecificEffect(RheumatoidArthritis)
- **Result**: Not supported. The context does not specify that the protein exclusively affects rheumatoid arthritis.
- **Step 3**:
- **Task**: Derivation fails.
- **Expression**: Derivation cannot proceed.
- **Result**: Cannot conclude that the medication does not affect other joint diseases.
- **Is Correct**: False
4. **Option Index**: 3

- **Option Text**: The benefits to rheumatoid arthritis sufferers of the new medication would outweigh the medication's possible harmful side effects.
- **Preconditions**:
- Cause(Hormone, Pain)
- Cause(Hormone, Swelling)
- Inhibit(Protein, Hormone)
- Contain(ArthritisMedication, Protein)
- **Deduction Target**: Benefit(ArthritisMedication) $>$ HarmfulSideEffect(ArthritisMedication)
- **Deduction Steps**:
- **Step 1**:
- **Task**: Identify the benefits of the medication based on inhibiting the hormone.
- **Expression**: Inhibit(Protein, Hormone) $\Rightarrow$ Reduce(Pain) $\wedge$ Reduce(Swelling)
- **Result**: Supported by context: The protein inhibits the hormone, which causes pain and swelling.
- **Step 2**:
- **Task**: Determine if the context provides information about harmful side effects.
- **Expression**: HarmfulSideEffect(ArthritisMedication)
- **Result**: Not supported. The context does not mention any side effects of the medication.
- **Step 3**:
- **Task**: Derivation fails.
- **Expression**: Derivation cannot proceed.
- **Result**: Cannot compare benefits and harmful side effects due to lack of information on side effects.
- **Is Correct**: False
Tips for Option Analysis
- **Preconditions**: Only include logical expressions that are directly relevant to the option being analyzed. Avoid listing all possible preconditions.
- **Deduction Steps**: Ensure each step logically follows from the previous one based on the preconditions. If a step cannot be completed due to insufficient support from the preconditions, indicate the failure and stop further deductions for that option.
- **is_correct**: This field should be true only for the option that matches the answer index. All other options should be false.
- **Format Consistency**: Maintain the same JSON structure and field naming conventions across all options to ensure uniformity and ease of data extraction.
- **Logical Accuracy**: Ensure that all logical expressions and deductions accurately reflect the relationships defined by the predicates and constants.
Your Task
Analyze the following Input data and generate the option_analysis section as per the example above. Replace the xxx placeholders in the example with actual data derived from the input.
Input Data
input_data_here
Please generate the option_analysis section based on the above input data.

---

## Prompt for GRPO training in BQA

You are solving a yes/no (BQA) problem. Follow these rules exactly: 1. First, think through your reasoning and enclose it in <think> and </think>. 2. Then, give your final answer ("yes" or "no") enclosed in <answer> and </answer>. 3. Do not output anything else; any deviation yields zero reward. 4. Your <think> block should not exceed 450 tokens. 5. Do not include <answer> inside your <think>, and vice versa. 6. Structure your reasoning in three steps in <think> block: Step 1: Analyze the Context. Step 2: Analyze the Question. Step 3: Deduce the Conclusion.
Example: Context: If Noah takes a vacation, then he will relax. If Noah works overtime, then he will not have enough time to relax. We know at least one of the following is true: he won't relax or he will have enough time to relax. We do not know which. Question: If at least one

of these statements is true, can we conclude that essentially "Noah takes a vacation" or "Noah works overtime" must be true?
Your response: <think>Step 1: Context Analysis: Let V="takes vacation", O="works over-time", R="relaxes", T="has enough time to relax". The context gives implications "if V is true, then R is true" and "if O is true, then T is false", and the disjunction "R is false" or "T is true". Step 2: Question Analysis: The question asks whether V or O must always be true given ?R or T. Step 3: Answer analysis: First, from "if V is true, then R is true", we can infer that "if R is false, then V is false". From "if O is true, then T is false", we can infer that "if T is true, then O is false". So, "R is false" or "T is true" implies that "V is false" or "O is false". Thus, V and O cannot both be true, and it's possible for both V and O to be false. Therefore, it's not certain that V or O must be true.</think><answer>no</answer>
Now solve this BQA problem:
Context:
Question:
Your response:

---

# D   DRI CALCULATION PROCESS AND RE-COGNIZING OPTIMIZATION ALGORITHM

Algorithm 1 illustrates the calculation process of our DRI score, and Algorithm 2 demonstrates the implementation procedure of our Re-Cognizing Optimization strategy.

**Algorithm 1** Data Reasoning Intensity Calculation

---

1: **Input:** A sample $x$ with context $c$ and options $\{o_l\}_{l=1}^L$.
2: Parse context $c$:
3:      Extract logical expressions $\mathcal{E}$ and compute their nesting depths.
4:      Identify predicates $\mathcal{P}$ and constants $\mathcal{C}$ in $c$.
5: Compute context intensity:
6:      $S_{\text{ctx}} = |\mathcal{E}| \times \bar{D}^2 + |\mathcal{P}| + |\mathcal{C}|$.
7: **for each** option $o_l$ **do**
8:      Extract preconditions $\mathcal{R}_l$ and reasoning steps $\mathcal{S}_l$.
9:      Compute option intensity:
10:      $S_{\text{opt}}^{(l)} = |\mathcal{R}_l| \cdot \bar{D}_l^2 + \sum_{k=1}^{T_l} \left(1 + \#\text{Operations}_{l,k}\right) D_{l,k}^2$.
11: **end for**
12: Aggregate raw intensity:
13:      $S_{\text{raw}} = S_{\text{ctx}} + \sum_{l=1}^L S_{\text{opt}}^{(l)}$.
14: Normalize to $[0, 1]$ via sigmoid of log:
15:      $S = \sigma\left(\gamma \cdot \dfrac{\log(S_{\text{raw}} + 1) - \mu}{\sqrt{\sigma^2 + \epsilon}} + \beta\right)$ .
16: **Output:** reasoning-intensity score $S$.

---

**Algorithm 2** Re-Cognizing Optimization

---

1: **Input:** dataset $\mathcal{D} = \{x_i\}_{i=1}^N$, model $M$, epochs $T$.
2: Precompute intensity scores $\{S_i\}_{i=1}^N$ via Algorithm 1.
3: **for** epoch $t = 1$ **to** $T$ **do**
4:      **if** $t = 1$ **then**
5:          Uniformly shuffle $\mathcal{D}$ for initial exploration (Phase I: Model Cognition Reshaping).
6:      **else**
7:          Sort $\mathcal{D}$ by descending $S_i$ to emphasize high-intensity samples (Phase II: Cognitive Reasoning Enhancement).
8:      **end if**
9:      **for** each batch $B$ drawn sequentially from $\mathcal{D}$ **do**
10:          Compute loss on $B$ and update model parameters.
11:      **end for**
12: **end for**
13: **Output:** fine-tuned model $M^*$.

---

# E    THEORETICAL FOUNDATIONS OF DRI

## E.1    DEFINITIONS OF CORE CONCEPTS

### E.1.1    INTRINSIC COGNITIVE COST $C(\mathcal{M})$

For a model $\mathcal{M}$, $C(\mathcal{M})$ represents the aggregate cost of its reasoning process, determined by a function of key model-intrinsic factors:

$$C(\mathcal{M}) = f(S, R, A)$$

Here, $S$ denotes model scale, which encompasses parameters size and the depth of transformer layers, collectively reflecting the baseline resource demand of the model. $R$ refers to reasoning computational complexity, specifically the number of operations required for executing logical deduction steps such as multi-step inference and symbolic manipulation. $A$ stands for architectural constraints, including design features like attention mechanisms and expert selection strategies that influence the efficiency of the reasoning process.

### E.1.2   DATA REASONING POTENTIAL $E(\mathcal{D})$

For a dataset $\mathcal{D}$, $E(\mathcal{D})$ quantifies the latent reasoning value embedded in its samples, defined as a function of critical data characteristics:

$$E(\mathcal{D}) = g(T, L, K)$$

$T$ represents structured reasoning traces, which are the step-by-step logical chains present in sample annotations. $L$ denotes logical component density, measuring the number of atomic reasoning units (such as causal inference and conditional judgment) per unit length of reasoning text. $K$ indicates semantic coherence, reflecting the consistency and relevance between consecutive reasoning steps within the dataset.

### E.2   DERIVATION OF EQUATION (1)

The effective reasoning capability $\eta(\mathcal{M}, \mathcal{D})$ is derived from the principle of resource-efficiency trade-off. The core logic underlying this derivation is that $E(\mathcal{D})$ embodies the reasoning information that can be exploited by the model, meaning higher $E(\mathcal{D})$ tends to promote better reasoning performance. Conversely, $C(\mathcal{M})$ reflects the resource consumption required for the model to complete reasoning, so higher $C(\mathcal{M})$ may limit the effective utilization of data potential.

Based on this relationship, we hypothesize that $\eta$ is positively correlated with $E(\mathcal{D})$ and negatively correlated with $C(\mathcal{M})$, leading to the proportional form:

$$\eta(\mathcal{M}, \mathcal{D}) \propto \frac{E(\mathcal{D})}{C(\mathcal{M})}$$

The equality in Equation (1) is established by normalizing this proportionality to a dimensionless metric, where the specific scaling coefficient is context-dependent, varying with different model types and dataset domains.

### E.3   BOUNDARY CONDITIONS

Equation (1) operates under the following implicit constraints:

- ***Compatibility Range:*** The model $\mathcal{M}$ must have a minimum capacity to process the dataset $\mathcal{D}$, i.e., $C(\mathcal{M}) \geq \kappa \cdot E(\mathcal{D})$, where $\kappa > 0$ denotes the minimal model-to-data capacity ratio required for meaningful reasoning. For highly mismatched pairs (such as lightweight models processing ultra-complex reasoning data), $\eta$ loses interpretability.

- ***Diminishing Returns:*** When $E(\mathcal{D})$ exceeds the reasoning boundary of $\mathcal{M}$—a threshold determined by the model's maximum processing capacity—the growth of $\eta$ slows down due to inherent capacity limitations.

- ***Interaction Effects:*** The separability of $E(\mathcal{D})$ and $C(\mathcal{M})$ is not absolute. Specialized models may exhibit higher $\eta$ for specific data structures, which is captured by context-dependent adjustments to the proportionality.

### E.4   THEORETICAL JUSTIFICATION OF DRI COMPONENTS

To validate the rationality of the proposed DRI metrics, we analyze the relationship between each component in the quantification formulas and the actual reasoning difficulty.

For the logical intensity score $S_{\text{ctx}}$:

- The term $|\mathcal{E}_i| \cdot \bar{D}^2$ reflects the structural complexity of reasoning. Logical expressions ($\mathcal{E}_i$) are the core carriers of reasoning logic, and their quantity directly affects the information processing load. The square of the average nesting depth ($\bar{D}^2$) is used because deeper nesting implies more nested logical operations (such as nested "if-then" structures), which exponentially increases the difficulty of parsing and deduction—consistent with the observation that complex structures in reasoning tasks create computational bottlenecks.

- The counts of predicates ($|\mathcal{P}_i|$) and constants ($|\mathcal{C}_i|$) capture the richness of the reasoning elements. More unique predicates mean more types of relationships need to be processed, and more constants increase the burden of entity mapping, both of which are basic factors affecting reasoning difficulty.

For the reasoning intensity score $S_{\text{opt}}^{(l)}$:

- The precondition term $|\mathcal{R}_l| \cdot \bar{D}_l^2$ quantifies the complexity of the initial logical assumptions. Similar to the structural term in $S_{\text{ctx}}$, it ensures that the difficulty of establishing preconditions (a key step in reasoning) is adequately reflected.

- The step-wise term $\sum_{k=1}^{T_l} (1 + \#\text{Operations}_{l,k}) \cdot D_{l,k}^2$ considers both the number of reasoning steps ($T_l$) and the complexity of each step. The operator count ($\#\text{Operations}_{l,k}$) directly measures the logical operations (AND/OR/NOT) involved, and the square of nesting depth ($D_{l,k}^2$) again emphasizes the impact of structural complexity—aligning with the intuition that each step's difficulty is determined by both its logical operations and structural depth.

