# OpenReview forum: "Pushing LLMs to Their Logical Reasoning Bound: The Role of Data Reasoning Intensity"
_ICLR.cc/2026/Conference — ICLR 2026 Conference Withdrawn Submission_

### Official Review · Reviewer_JAg8 · 2025-10-29

**Soundness:** 2
**Presentation:** 2
**Contribution:** 2
**Rating:** 4
**Confidence:** 4

**Summary:**

This paper introduce Data Reasoning Intensity (DRI), a novel metric that quantifies the latent logical reasoning complexity of samples by decomposing and aggregating their logical structures.
Based on this, the authors introduce a re-cognizing optimization strategy that systematically enhances the logical reasoning intensity of training data.
Experiments show that the proposed approaches improves performance and generalization over data-centric strategies.

**Strengths:**

1. The paper develops a quantitative metric for the reasoning complexity of training data.
2. The authors draw on established theories like  Sweller’s cognitive load optimization principle to propose the re-cognizing optimization strategy.

**Weaknesses:**

1. Although the paper introduces a quantitative metric for data reasoning potential, the rationale behind the specific formulations (e.g., Equations 1, 5, 6, and 7) is unclear: why are they defined in their current forms rather than other possible ones (linear, polynomial, exponential, logarithmic, etc.)? The paper lacks sufficient theoretical justification, intuitive explanation, or empirical validation for these choices.
2. The definition in Equation 1 is somewhat confusing. It is unclear why it is termed effective reasoning capability: it seems the authors may intend to express the model’s efficiency or learning ability in utilizing data per parameter unit. Moreover, since model architectures may naturally exhibit different sensitivities to various data types, it is not clear how the coupling between C(M) and E(D) is considered.
3. The notations used in Sections 3.1 and 3.2 are also difficult to follow. It would be helpful if the authors could include concrete examples to illustrate the meaning or numerical form of these symbols.
4. Figure 1 lacks sufficient explanation. The meaning of the horizontal gray dashed line and the definition of the ideal state are unclear. According to the intended meaning of Equation 1, shouldn’t the line be linear? (Though I may have misunderstood the authors’ intention.)
5. Beyond natural-language-based logical reasoning, mathematics and code are important components of reasoning tasks today. The generalization and applicability of the proposed metric to these domains require further validation.

**Questions:**

see Weakness.

---

> ### Author Response · Authors · 2025-11-15
>
> Thank you for the reviewer’s careful reading and constructive feedback. We provide detailed clarifications and responses to the key concerns below.
>
> ---
> 1. On the motivation and rationality behind Eq. (1), (5), (6), and (7)
>
> Eq. (1) is inspired by the Roofline Model (Williams et al., 2009) and is used to illustrate that an LLM’s reasoning efficiency is jointly constrained by two factors: the data-side reasoning potential ($E(D)$) and the model-side cognitive cost ($C(M)$).
>
> We emphasize that Eq. (1) is a motivational theoretical framework rather than a definition or computational formula for DRI. Its role is to explain why improving data-side reasoning structure is a practical and effective way to enhance model reasoning capability.
>
> The actual definition of DRI comes from Eq. (5)–(7), which explicitly decompose each sample’s structured reasoning trace into quantifiable logical components, including the number of expressions, predicate/constant density, squared expression depth, operator complexity, and multi-step reasoning structure. These components correspond to genuine contributors to reasoning difficulty.
>
> Across multiple logical reasoning benchmarks, we empirically evaluated different formulations and found that the current design exhibits stability, discriminative power, and natural correlation with reasoning error rates. Thus, Eq. (5)–(7) reflect a combination of theoretical motivation and empirical validation, rather than arbitrary choices.
>
>
> ---
> 2. On the meaning of “effective reasoning capability” in Eq. (1), and the relationship between ($C(M)$) and ($E(D)$)
>
> We fully understand the reviewer’s concern about Eq. (1). “Effective reasoning capability” is intended to express that a model’s reasoning performance is jointly constrained by:
>
> 1. $E(D)$: the reasoning demand encoded in the dataset, including compositional complexity, multi-step inference depth, and symbolic abstraction;
>
> 2. $C(M)$: the model’s intrinsic cognitive cost, encompassing architectural capacity, parameter scale, and reasoning FLOPs, which together shape the model’s inherent reasoning burden.
>
> Thus, Eq. (1) provides a macro-level efficiency perspective, whereas DRI specifically operationalizes and quantifies ($E(D)$).
>
> Regarding the reviewer’s concern that “different architectures may have different sensitivities to data,” we fully agree. As discussed in Appendix E.3 on Interaction Effects, ($E(D)$) and ($C(M)$) are not strictly separable; different architectures may utilize the same reasoning structure with varying efficiency. This is precisely why we validated DRI on two architecturally distinct models, LLaMA and Qwen, and observed consistent benefits, suggesting that DRI remains stable across different cognitive structures.
>
>
> ---
> 3. On the abstraction level of the formulas
>
> We thank the reviewer for pointing out the high abstraction level of some symbols.
>
> In the appendix, we provide extended explanations of all symbols, logical decomposition steps, and structured-chain construction. In the revised version, we will add a concrete example demonstrating how a logical sentence is mapped to its predicate set, constant set, expression depth, and structural components, to further enhance clarity.
>
>
> ---
> 4. On the interpretation of Figure 1 and the meaning of the dashed line
>
> We appreciate the reviewer’s attention to Figure 1. The figure is intended as a conceptual illustration, not an exact functional relationship. It conveys that model performance is limited by both reasoning potential and cognitive cost. The dashed gray line represents the ideal upper bound assuming perfect utilization of reasoning potential. In practice, model capacity limits and attention bottlenecks lead to nonlinear performance trends, so the plot is not meant to represent a precise functional mapping.
>
> We will explicitly label Figure 1 as a conceptual illustration in the revision to avoid confusion.
>
>
> ---
> 5. On the generalization of DRI to mathematics and code reasoning
>
> We fully agree that mathematics and code constitute essential aspects of modern reasoning research.
>
> Our current work focuses on natural-language logical reasoning because its structured reasoning traces make it well-suited for validating DRI’s decomposition framework. Due to space and compute constraints, we could not include cross-domain experiments in the present submission. That said, DRI’s structure-based formulation is modality-agnostic by design. We plan to extend it to mathematical reasoning and code reasoning to further evaluate its cross-modal applicability.
>
> ---
>
> We thank the reviewer again for raising these important points. The feedback helps us strengthen the theoretical motivation, formula interpretation, symbolic clarity, and illustration explanation. We look forward to any further suggestions from the reviewer.

---

### Official Review · Reviewer_MCeZ · 2025-11-01

**Soundness:** 2
**Presentation:** 2
**Contribution:** 2
**Rating:** 2
**Confidence:** 3

**Summary:**

The authors suppose that the LLM's reasoning ability is constrained by the potential of the training data. Therefore, the authors design a metric, named data reasoning intensity, to re-order the training datasets. In this way, the LLMs are trained on re-ordered datasets from low DRI samples to high DRI samples. Experiment results prove the effectiveness of the proposed method. The DRI depends on the Logical Intensity Quantification and Reasoning Intensity Quantification.

**Strengths:**

1. The paper proposes a new metric to re-range the training samples of reasoning tasks.
2. The re-range dataset is balanced. According the Figure 3 (c), a high DRI score also means a higher error rate, which is consistent.
3. The fine-tuned models show better performance on reasoning tasks.

**Weaknesses:**

1. The improvements are not significant compared to baselines.
2. In Figure 1, the 8b-our method is closed to the 30b-base. However, there are no related experiments and results in Section 4.

**Questions:**

Does this method improve an 8B model to achieve 32B-level performance?

---

> ### Author Response · Authors · 2025-11-15
>
> We sincerely thank the reviewer for the thoughtful evaluation and constructive comments. Below we provide clarifications regarding the concerns on performance gains and the interpretation of Figure 1.
>
> ---
> 1. On the concern that the improvements are not significant
>
> We agree that the performance gains reported in some benchmarks are not large. This mainly relates to the core purpose of our study:
> to validate the effectiveness and operability of DRI as a quantitative measure of data reasoning potential, rather than to solely maximize numerical improvement.
>
> To ensure fair comparison, we deliberately constrained both the dataset size and model scale, so that different methods are evaluated under the same controlled setting. Under such constraints, the performance ceiling of the model is naturally limited by the available data and the inherent difficulty of the tasks. Even modest improvements are meaningful because:
>
> 1. DRI consistently brings positive and stable gains across all benchmarks, in both SFT and RL settings;
>
> 2. Results show that DRI effectively identifies high-value samples and improves the model’s ability to learn from high-intensity reasoning data;
>
> 3. The improvement trends are uniform across multiple logical reasoning datasets, offering statistical support for the soundness of DRI.
>
> Thus, although the improvement margins are moderate, the experiments substantially validate the value and transferability of modeling data reasoning potential, laying the groundwork for future data-centric reasoning optimization.
>
> ---
> 2. On the question regarding Figure 1 and the “8B-our method close to 30B-base” statement
>
> Thank you for pointing out this issue. We clarify that Figure 1 is a conceptual illustration, not an empirical result.
>
> Its purpose is to convey the motivation of our framework: a model’s reasoning performance is jointly constrained by the data-side reasoning potential ($E(D)$) and the model’s intrinsic cognitive cost ($C(M)$), and improving data reasoning intensity can help a model approach its ideal performance upper bound.
>
> Therefore, the visual impression that “8B-our method is close to 30B-base” is not intended to claim that our method enables an 8B model to reach the actual performance of a 30B model. Instead, it illustrates that:
>
> 1. Under the same dataset, enhancing data reasoning intensity allows the smaller model to approach the performance upper bound corresponding to that reasoning regime;
>
> 2. This conceptual relationship motivates our proposed Re-Cognizing Optimization, which restructures data reasoning intensity so that smaller models can utilize data more efficiently.
>
> In the revision, we will explicitly indicate in the caption that Figure 1 is a conceptual diagram to avoid misunderstanding.
>
>
> ---
> We thank the reviewer again for the valuable comments. While the improvements are moderate, the essence of our work is to introduce a principled framework for quantifying reasoning potential in training data and to reveal a measurable relationship between data quality and model cognitive ability. We believe this perspective provides useful tools for future data-driven reasoning enhancement, and we welcome further feedback from the reviewer.

---

### Official Review · Reviewer_uzRr · 2025-11-03

**Soundness:** 2
**Presentation:** 3
**Contribution:** 3
**Rating:** 4
**Confidence:** 4

**Summary:**

This paper introduces Data Reasoning Intensity (DRI), a novel metric for quantifying the latent logical reasoning complexity of training samples for large language models. The authors posit that LLM reasoning performance is jointly constrained by training data potential and model cognitive capacity. DRI decomposes samples into logical elements (predicates, constants, expressions) and aggregates them via a scoring function that accounts for structural complexity, nesting depth, and reasoning steps. Based on empirical analysis showing current LLMs underutilize available reasoning data, the authors propose a re-cognizing optimization strategy: Phase I reshapes model cognition through uniform data exposure, while Phase II emphasizes high-DRI samples via probability-weighted sampling. Experiments on four logical reasoning benchmarks (Reclor, LogicBench, LogiQA, LogiQA2.0) using LLaMA3.1-8B and Qwen2.5-7B demonstrate consistent improvements over curriculum learning and bin-based progressive learning baselines in both supervised fine-tuning and reinforcement learning settings.

**Strengths:**

- The paper includes extensive experiments across multiple datasets, model architectures, and training paradigms (SFT and RL), with careful construction of balanced test sets to avoid evaluation bias.
- The analysis reveals three important and clear observations: (1) low-DRI data can be pruned, (2) high-DRI data catalyzes improvements, and (3) diversity across DRI spectrum is necessary—providing actionable guidance for data curation.
- The connection to the Roofline Model and cognitive load theory provides intuitive motivation, even if the formalization could be strengthened.

**Weaknesses:**

- The quantity DRI well supports the data selection given the model complexity. However, this is how E(D) part works. If the author wants to advocate that DRI works as the ratio of E(D) / C(M), as the key contribution, the C(M) part should also be demonstrated.
- The E(D) quantity is defined on logical problems, specifically, a well defined first order language. This aligns well with the scope of this paper. However, one potential limitation is that if we work on a first order language where $|C_i|$ is particularly large, say a database on all athletics in Olympic games from 1900. where the |C_i| term dominate the calculation so that the DRI score can be biased. This can happen if LLM is given a question about athletics and a full list of their names are input in the context. How will you define the E(D) score for the sample?

**Questions:**

- Can we say that, the DRI ratio E(D) / C(M) means that more parameters of LLM leads to lower DRI score given the same dataset. So can we say, for a 30B model and a 8B model, if we train it on the same dataset D, the 30B model should perform worse than the 8B one because its DRI is low?

---

> ### Author Response · Authors · 2025-11-15
>
> We sincerely thank the reviewer for the careful reading and constructive feedback. Below we clarify the key issues raised in the review.
>
> ---
>
> ### 1. Clarification on the relationship between Equation (1) and DRI
> Thank you for pointing out the expression
> $$\eta(M, D) = \frac{E(D)}{C(M)}.$$
>
> We would like to clarify that **Equation (1) is purely motivational and is not used to compute DRI**.
>
> Its purpose is to provide a conceptual framework inspired by the Roofline Model, illustrating that a model’s effective reasoning capability is jointly constrained by:
> 1. The reasoning potential of data ($E(D)$);
> 2. The model’s cognitive cost ($C(M)$).
>
> In our actual method, **DRI depends solely on $E(D)$**. It is an operational approximation based on logical decomposition and aggregation (Section 3, Equations (5)–(8)). DRI is *not* the ratio $E(D)/C(M)$.
> The term $C(M)$ serves only as a theoretical motivation and is not a tunable variable in our method.
>
> We will clarify this distinction between motivation and definition in the revision.
>
> ---
>
> ### 2. On the potential issue that a large constant set ($|C_i|$) may dominate the score
> We understand the reviewer's concern. In our design:
> 1. $|C_i|$ is only one component among structural complexity, predicate–constant distribution, and reasoning-depth quantification (Equations (6)–(8));
> 2. Log and sigmoid normalization compress the influence of large constant sets;
> 3. Preprocessing removes irrelevant constant-heavy text (e.g., long entity lists).
>
> Empirically (Figure 3(c)), we did not observe instability caused by large $|C_i|$, indicating the score remains balanced.
>
> ---
>
> ### 3. On whether a 30B model would perform worse than an 8B model because DRI becomes lower
> Thank you for raising this question. The misunderstanding originates from interpreting Equation (1) as the definition of DRI.
> 1. **DRI is a property of data only and is independent of model size.**
> 2. $C(M)$ denotes the model’s intrinsic cognitive cost, encompassing architectural capacity, parameter scale, and reasoning FLOPs.
>
> Larger models usually have stronger abstraction and representation ability, so:
> $$\text{larger model size} \Rightarrow \text{lower } C(M)$$
>
> meaning that a 30B model can utilize the same data reasoning potential ($E(D)$) more effectively than an 8B model.
>
> Thus, a larger model performs better not because DRI changes (it does not), but because $C(M)$ is lower.
>
> ---
>
> We sincerely thank the reviewer again for the insightful comments. These points help us clarify the distinction between the motivational ratio in Equation (1), the practical definition of DRI, and the robustness of DRI under extreme conditions. We will revise the manuscript accordingly and would greatly appreciate further feedback.

---

### Note · Authors · 2025-12-02

I have read and agree with the venue's withdrawal policy on behalf of myself and my co-authors.